# Chromosome Rearrangement in *Elymus dahuricus* Revealed by ND-FISH and Oligo-FISH Painting

**DOI:** 10.3390/plants12183268

**Published:** 2023-09-14

**Authors:** Chengzhi Jiang, Xiaodan Liu, Zujun Yang, Guangrong Li

**Affiliations:** School of Life Science and Technology, University of Electronic Science and Technology of China, Chengdu 610054, China; 201921140403@std.uestc.edu.cn (C.J.); 202021140504@std.uestc.edu.cn (X.L.)

**Keywords:** *Elymus dahuricus*, ND-FISH, Oligo-FISH painting, chromosome rearrangement

## Abstract

As a perennial herb in Triticeae, *Elymus dahuricus* is widely distributed in Qinghai–Tibetan Plateau and Central Asia. It has been used as high-quality fodders for improving degraded grassland. The genomic constitution of *E. dahuricus* (2n = 6x = 42) has been revealed as StStHHYY by cytological approaches. However, the universal karyotyping nomenclature system of *E. dahuricus* is not fully established by traditional fluorescent in situ hybridization (FISH) and genomic in situ hybridization (GISH). In this study, the non-denaturing fluorescent in situ hybridization (ND-FISH) using 14 tandem-repeat oligos could effectively distinguish the entire *E. dahuricus* chromosomes pairs, while Oligo-FISH painting by bulked oligo pools based on wheat-barley collinear regions combined with GISH analysis, is able to precisely determine the linkage group and sub-genomes of the individual *E. dahuricus* chromosomes. We subsequently established the 42-chromosome karyotype of *E. dahuricus* with distinctive chromosomal FISH signals, and characterized a new type of intergenomic rearrangement between 2H and 5Y. Furthermore, the comparative chromosomal localization of the centromeric tandem repeats and immunostaining by anti-CENH3 between cultivated barley (*Hordeum vulgare* L.) and *E. dahuricus* suggests that centromere-associated sequences in H subgenomes were continuously changing during the process of polyploidization. The precise karyotyping system based on ND-FISH and Oligo-FISH painting methods will be efficient for describing chromosomal rearrangements and evolutionary networks for polyploid *Elymus* and their related species.

## 1. Introduction

*Elymus* L. is mainly distributed in Eurasia and North America, which can be used as high-quality fodders and effective plant tools for restoring degraded field [1,2]. *Elymus* L. is a large genus in the tribe Triticeae (Poaceae), containing approximately 150 perennial and polyploid species with their genomes designated as StH, StY, StHY, StPY, and StWY [3,4,5]. Based on the description of numerous tetraploid and hexaploid species, *Elymus* L. can offer an appropriate model to investigate the dynamic processes of the genome in recurrent formation of polyploid under contrasting environmental factors. As a sparse and perennial herb, *Elymus dahuricus* is highly adapted to the harsh natural environments in alpine regions, and the genome constitution of *E. dahuricus* is described as StStHHYY (2n = 6x = 42) [6]. The basic genomes St, H and Y in *E. dahuricus* are derived from *Pseudoroegneria*, *Hordeum* and an unknown ancient species, respectively [7].

Chromosomal rearrangements (CRs) including translocations, inversion and deletions, play significant roles in maintaining genome stability and karyotype evolution during the process of speciation [8,9]. CRs have been frequently observed in normal diploid and polyploid Triticeae species [10], and the formation of CRs can be detected by distant hybridization between wheat and relative species of Triticeae [11]. Besides, a high frequency of CRs in wheat can also be induced by radiation [12,13]. Above all, the hypothesis of nucleo-cytoplasmic interaction (NCI) explains that species-specific CRs play a strong part in restoring fertility and nucleo-cytoplasmic compatibility in newly formed allopolyploid, which have been investigated in *Elymus* species among different subgenomes [14]. Further new molecular cytological techniques for the determination of novel species-specific CRs will be helpful for revealing the evolutionary aspect of polyploid *Elymus* species.

Fluorescence in situ hybridization (FISH) and genomic in situ hybridization (GISH) are powerful for the construction of karyotypes and identification of CRs in Triti-ceae species [15,16]. A low-cost non-denaturing FISH (ND-FISH) technology, using oligonucleotide (oligo) probes representing repetitive sequences, can effectively analyze the genetic diversity involved in chromosome structural variation among different species or accessions [17,18]. For instance, a high-resolution ND-FISH karyotype of *Roegneria ciliaris* was established and different types of CRs were detected among 53 *R. ciliaris* accessions from various ecological regions [19]. Moreover, Oligo-FISH painting system developed by seven bulked pools, can successfully assign the individual chromosome to specific linkage groups and identified many CRs in several Triticeae species [20]. In previous studies, the genomic constitution of *E. dahuricus* was analyzed by FISH of 5S rDNA sequences, and intergenomic rearrangements between H and Y genomes were observed by GISH [6,21]. However, the studies for revealing complicated genomic changes have not been established due to the lack of universal karyotyping nomenclature system for each individual *E. dahuricus* chromosome pair.

In the current study, ND-FISH using 14 tandem-repeat oligos and Oligo-FISH painting based on linkage specific oligo pools were employed to establish a standard nomenclature system for the classification of individual chromosomes of *E. dahuricus*. Meanwhile, we characterized a novel intergenomic rearrangement between 2H and 5Y, and further found that the centromere-associated sequences in H subgenome have undergone expansion and rearrangement during polyploidization of *E. dahuricus*. The present study has provided useful tools to characterize chromosomal rearrangements and karyotypic diversity for polyploid *Elymus* species.

## 2. Results

### 2.1. Genome Identification Using Combined FISH and GISH Analyses

GISH analysis using total genomic DNA from *Pseudoroegneria spicata* (St genome) as a probe on metaphase chromosomes of *E. dahuricus*. Seven pairs of chromosomes have strong green signals (Figure 1a), indicating that *E. dahuricus* contained St genome. Afa-family repetitive sequences had enriched hybridization sites in each H chromosomes of *Elymus* species [22]. Subsequently, the ND-FISH by Afa-family representing probe Oligo-pTa535 combined with barley centromeric sequences Oligo-HvCSR was used to characterize the H genome. As shown in Figure 1b, the ND-FISH results showed eight pairs of chromosomes with strong Oligo-HvCSR hybridization sites at the centromeric region. Meanwhile, seven pairs of H genome showed strong hybridization signals of Oligo-pTa535 in pericentric, interstitial, or subtelomeric regions, which clearly distinguished the H genome from those of the St and Y genomes (Figure 1b). Combination of ND-FISH and GISH analysis suggested that seven pairs of H-chromosomes could be characterized by Oligo-pTa535, and centromeric repeats may be rearranged between H and St genomes (Figure 1b).

The remaining seven pairs of chromosomes absent in St-GISH and Oligo-HvCSR signals belonged to Y genome. These Y-chromosomes demonstrated the slightly shorter than St and H-chromosomes in *E. dahuricus* (Appendix A). Thus, we were able to distinguish the St, Y and H subgenomes in *E. dahuricus*, and further multiple oligo probes for ND-FISH were added to characterize 21 pairs of *E. dahuricus* chromosomes.

### 2.2. Precise Chromosome Identification by ND-FISH

After distinguishing the different subgenomes in *E. dahuricus*, the sequential ND-FISH using 14 Oligo probes were added to precisely construct the karyotype of the complete 42 chromosomes (Table 1). Seven of them were newly synthesized Oligo probes, namely Oligo-P05, Oligo-13-J1011, Oligo-D01-135, Oligo-HvCSR, Oligo-Ae334, Oligo-7E-716 and Oligo-7E-599, which were designed from the sequenced genome of Triticeae species by the method of Lang et al. [23]. The rest of probes Oligo-pSc119.2, Oligo-pTa535, Oligo-5SrDNA, Oligo-3A1, Oligo-18SrDNA, Oligo-7E-744, and Oligo-(GAA)_7_ are previously reported.

Each mitotic mataphase chromosomal slide of *E. dahuricus* was first analyzed by ND-FISH with probe combinations of Oligo-pTa535 + Oligo-pSc119.2 and seven pairs of H-chromosoms showed distinct hybridization signals (Figure 2a,d,g). Three Oligo-5SrDNA sites, each in the H, Y and St genomes, and three Oligo-18SrDNA sites, of which two in the St genomes and one in the Y genome, were detected in *E. dahuricus* (Figure 2b). Meanwhile, the polymorphic FISH signals of Oligo-5SrDNA and Oligo-18SrDNA sites were observed in St genome (Appendix A). As shown in Figure 2c, the probe Oligo-13-J1011 produced different hybridization signals on centromeric or pericentromeric regions of five pairs of Y-chromosomes, as for remaining two pairs of Y-chromosomes, one carried strong Oligo-P05 signal at telomeric regions, and one demonstrated the large arm ratio. Therefore, in combination with Oligo-13-J1011 and Oligo-P05 probes, the Y chromosomes in *E. dahuricus* were clearly recognizable. Similarly, the probes Oligo-Ae334 + Oligo-D01-135, displayed distinguishing hybridization patterns in Y genome, as well as characterized four pairs of chromosomes in St genome (Figure 2h). Besides H chromosomes, Oligo-HvCSR showed strong centromeric signal on one pairs of chromosomes in St genome, and probe Oligo-7E-599 produced intensive hybridization signals on telomeric or subtelomeric regions of one or both arms of most St and Y-chromosomes. Meanwhile, probe Oligo-7E-716 had strong hybridization signals at short arms of two Y-chromosomes, and weaker signals at long arms of three St chromosomes, and Oligo-7E-744 signals were also observed on telomeric regions of three St chromosomes (Figure 2e,f). Seven pairs of St chromosomes can be recognized by the different combination of these four probes. Furthermore, six chromosomes in the H genome and five chromosomes in the St genome produced Oligo-(GAA)_7_ signals, but only one chromosome did so in the Y genome (Figure 2i). Finally, we found two H-chromosomes showed distinct hybridization sites using probe Oligo-3A1, the other two sites appeared in St and Y genomes (Figure 2i). Therefore, different chromosomes in each subgenome have distinguishing hybridization patterns, and 21 pairs of chromosomes in *E. dahuricus* could be precisely identified by multiple oligo probes. However, the assignment of linkage group in *E. dahuricus* were further precisely determined by Olig-FISH painting techniques.

### 2.3. Identification of Chromosome Linkage Groups

After recognizing 21 pairs of chromosomes in *E. dahuricus* by ND-FISH using 14 Oligo probes, the sequential Oligo-FISH painting with bulked Oligo probes Synt1 to Synt7 were used to determine the linkage group of each *E. dahuricus* chromosome through signal distribution and specificity.

In this study, we selected six probes Oligo-pTa535, Oligo-pSc119.2, Oligo-13-J1011, Oligo-7E-744, Oligo-D01-135 and Oligo-(GAA)_7_ to characterize all chromosomes as first round of ND-FISH analysis (Figure 3a,c,e,g,i), and the sequential Oligo-FISH painting with different oligo pool probes combinations were used for the second round of hybridization (Figure 3b,d,f,h,j). For example, FISH painting with probe Synt6 revealed that along the entire lengths of six chromosomes of three homoeologous pairs were strongly hybridized by red signals. They belonged to each St, Y and H subgenome, and thus designated as 6St, 6Y and 6H combined with the ND-FISH results. Also the probe Synt4 produced distinct green signals on three chromosomes pairs of 4St, 4Y and 4H (Figure 3b). Similarly, the oligo pool probes Synt1, Synt3 and Synt7 were also hybridized to the *E. dahuricus* chromosomes, and produced distinct signals on all lengths of three homoeologous chromosomes pairs of the linkage groups 1, 3 and 7, respectively (Figure 3f,h,j). In contrast, the complex hybridization patterns with probes Synt2 and Synt5 were detected. For instance, probe Synt2 generated clear green signals on four pairs of chromosomes (Figure 3f). Two pairs of chromosomes showed the entire lengths of green signals, and characterized as 2Y and 2St by ND-FISH. While, the Synt2 signals covered partial regions of remaining two pairs of chromosomes 2H-1 and 2H-2. Synt5 also displayed the similar signals patterns as those of Synt2 (Figure 3d), which is identical to the ND-FISH for the CRs in 5Y chromosomes. Therefore, the results suggested that a non-homologous CRs occurred in 2H and 5Y chromosomes of *E. dahuricus*.

Subsequently, painting probes Synt2 and Synt5 combined with ND-FISH were used to further characterize the break point of novel CRs in *E. dahuricus* (Figure 4). The Oligo-FISH painting confirmed the rearrangement of chromosomes 2HS.5YL and 5YS.2HL (Figure 4b,c), and the ND-FISH on 2HS.5YL and 5YS.2HL showed that the breakpoint at the sub-telomeric section of 2HS was close to the location of strong Oligo-pTa535 signals, while the breakpoint of 5YS was close to the weaker Oligo-pTa535 signals (Figure 4a,c).

### 2.4. Establishment of Universal Karyotyping Nomenclature System

Based on the previous ND-FISH patterns of 21 pairs of chromosomes with multiple probes and the assignment of 1 to 7 linkage groups in each H, St, Y subgenomes by Oligo-FISH painting, we are were to establish the standard karyotypes of *E. dahuricus* (Figure 5a). The ideogram for three genomes chromosomes of *E. dahuricus* are showed by the hybridization patterns of different probes (Figure 5b). It is clearly indicated that the abundance of repeats sequences of Oligo-pTa535 distributed across H genome. The Y and H-chromosomes showed more hybridization signals than St-chromosomes, implying that the St genome may have less distribution of tandem repeats than those of Y and H genomes. In contrast, the shortest chromosome 4St showed only two hybridization sites by oligo probes Oligo-P05 and Oligo-HvCSR. Thus, the universal karyotyping nomenclature system of individual *E. dahuricus* chromosomes was established, which can be extremely useful for analyzing chromosomal rearrangements, as well as revealing the genomic diversity in polyploid *Elymus* species.

### 2.5. Comparison Analysis of Centromere Specific Repeat Sequences and CENH3

In order to compare the location of centromeric protein and the distribution of centromeric repeat sequences of H genome between diploid barley and hexaploid *E. dahuricus* species, immuno-FISH analysis of anti-CENH3 and ND-FISH of Oligo-HvCSR were conducted. All chromosomes had the CENH3 signals in the centromeric regions, implying that CENH3 protein variants are deposited in the centromeric position among St, Y and H subgenomes (Figure 6a,d). Meanwhile, the weakly detectable signals in hexaploid species demonstrated that the accumulation of CENH3 on H chromosomes in E. dahuricus was less than that in barley (Appendix A). Sequential ND-FISH using probes Oligo-(GAA)_7_ and Oligo-pTa535 + Oligo-pSc119.2 confirmed identification of the H chromosomes in the same metaphase cells in barley and *E. dahuricus*, respectively (Figure 6c,f). We further estimated the average relative fluorescence intensity of CENH3 for each H chromosome, and found a rapid increase of relative fluorescence intensity of 6H in *E. dahuricus* and decrease in barley (Figure 6g).

Barley centromere-specific repeat Oligo-HvCSR probe produced strong hybridization signals in all H genomes (Figure 6b,e). The measurement of average relative fluorescence intensity of Oligo-HvCSR signals (Appendix A) revealed a significant increase of signal intensity detected in 6H chromosome (Figure 6h). Besides, chromosomes 4St and 6St of *E. dahuricus* displayed distinct and faint Oligo-HvCSR signals, respectively (Figure 6c,f), indicating that the rearrangements of centromeric repeats occurred between H and St genomes in polyploidization. Thus, it is suggested that the rate of evolution in centromeric specific repeats among each H chromosome is independent, while the chromosome 6H may exhibit a high variation during polyploidization.

The chromosome 2HS.5YL exhibited strong Oligo-HvCSR signals at the intercalary region of the short arm, which has different location with CENH3 immunostaining (Figure 6i). The absence of Oligo-HvCSR signals in another rearranged chromosome 5YS.2HL indicated that the centromeric specific repeat sequences and centromeric functional CENH3 region on 2H chromosome may transfer to 2HS.5YL at the beginning of CRs formation, and a new centromere was repositioned. Furthermore, we found the distinct signals of probe Oligo-HvCSR at the proximal region of the long arm on 1H, 4H, 4St and 6St (Figure 6i). The results indicated that the centromere-specific sequences of H genome tend to spread along the chromosome long arms, which may have caused centromere expansion during the process of polyploidization.

## 3. Discussion

Genomic in situ hybridization (GISH) and fluorescence in situ hybridization (FISH) are the most efficient techniques to process the cytogenetic studies in bread wheat and related species [28]. Based on GISH pattern using *Pseudoroegneria* and *Hordeum* genomic DNA (St and H genome) as probes, the genomic constitution of *E. dahuricus* was confirmed as StYH [6,14], and further sequential FISH using Afa-family sequences produced distinguishing signals in H genome [22]. However, precise identification of individual chromosomes and linkage groups of *E. dahuricus* is still limited by traditional FISH and GISH. Li et al. [20] developed seven bulked pools for an Oligo-FISH painting system and successfully characterized the non-homologous chromosomal rearrangements in *Secale*, *Aegilops* and other Triticeae species. Recently, single-gene FISH mapping was applied to identify intra-genome translocations and inversions in *E. sibiricus* and *E. nutans* [29]. The combination of oligo painting and single-gene FISH techniques will still be precisely identify the chromosomal structural variations in *Elymus* species as well as Triticeae species. In our study, we established the 42-chromosomes karyotyping of *E. dahuricus* with the precise assignment of subgenomes and linkage groups based on the Oligo-FISH painting and ND-FISH with multiple probes. The universal karyotyping nomenclature system can be used to describe the genetic diversity and chromosomal rearrangements among *Elymus* species.

Chromosomal rearrangements (CRs) play an important role during polyploidization and karyotypical evolution [9,10], which have been frequently observed in wheat and other Triticeae species [11,20]. For example, high-resolution multiple oligonucleotide FISH revealed 14 structural rearrangements from 373 Chinese wheat cultivars [30]. FISH using D-specific oligo probes characterized specific CRs involved 3D-7D and complex 4D-5D-7D in *Aegilops tauschii* from different origins [31]. Meanwhile, Li et al. [20] developed seven oligonucleotide pools for Oligo-FISH painting techniques, and demonstrated the multiple CRs for 3R, 4R, 6R and 7R occurred in rye. Chen et al. [19] identified reciprocal translocation 5YcS-5ScS.5ScL, 5ScS-5YcS.5YcL and 5YcS-6ScS.6ScL, 6ScS-5YcS.5YcL in *R. ciliaris* by FISH using three oligo probes. In *Elymus* species, intergenomic CRs between H/Y genome were identified in *E. xiningensis*, *E. dahuricus*, *E. barystachyus* and *E. excelsus* by mc-GISH [6], and a novel CRs between H and St genomes was also observed in *E. tangutorum* [14]. The absence of St/Y translocations may be possibly be attributed to the close homology between St and Y genomes [32]. However, the precise identification of CRs in *Elymus species* is still limited due to traditional FISH and GISH. In recent study, Liu et al. [29] developed single-gene FISH probes from the cDNA of *Elymus* species, and characterized eight species-specific CRs in *E. sibiricus*, including seven inversions in 1H, 2H, 3H, 6H, 2St, 4St and 5St; one reciprocal 4H/6H translocation, five species-specific CRs were identified in *E. nutans*, including four inversions in 1H, 2H, 2Y and 4Y, and one reciprocal 4Y/5Y translocation. In the present study, we found an intergenomic CRs between 2H and 5Y chromosomes in *E. dahuricus*, namely 2HS.5YL and 5YS.2HL, and further revealed that the rearrangements occurred in the middle of short arms of each chromosome by ND-FISH and Oligo-FISH painting using Synt2 and Synt5 probes. It was reported that *E. dahuricus* complex and *E. excelsus* share the same H/Y reciprocal translocation and show the similar FISH and GISH patterns [14]. It is important to note that the species-specific genomic translocation existed in *Elymus* species, which may drive the speciation and adaptation for their specific grown environment [33].

The 18S rDNA and 5S rDNA sequences have been used as FISH probes to study the genetic and evolutionary relationships in wheat and related species [34,35]. In previous studies, *E. dahuricus* displayed two 18Sr DNA-5Sr DNA sites in the St genome and one site in Y genomes [6]. However, our study shows that the distinct 5Sr DNA signals are located on the short arm of 5H, 1St and 7Y, and strong 18Sr DNA signals are distributed in the short arm of 1Y, 1St and 5St, as well as the rearrangement chromosome 5YS.2HL of *E. dahuricus* (Appendix A), suggesting that the 18S rDNA and 5S rDNA are conserved in homoeologous 1 and 5 groups in *E. dahuricus* species. In previous study, Taketa et al. [36] characterized the hybridization patterns of 18S rDNA-5S rDNA sites in different wild *Hordeum* species for example, *Hordeum bulbosum* (2n = 14; I-genome) had a pair of 5S rDNA sites and a pair of 18S rDNA sites on different chromosomes; *Hordeum chilense* (2n = 14; H-genome) carried an double 5S rDNA site on short arm of 5H^ch^ and two pairs of 18S rDNA sites on 5H^ch^ and 6H^ch^; *Hordeum marinum* (2n = 14; X-genome) displayed a pair of 18S rDNA sites and two pairs of 5S rDNA sites; *Hordeum glaucum* (2n = 14; Y-genome) and Hordeum murinum (2n = 14; Y-genome) both had two pairs of 18S rDNA sites and a pair of 5S rDNA sites. In our study, we analyzed the Oligo-5S rDNA signals of H genome in barley, and found that a strong hybridization signal in 2H, and three weaker signals on 3H, 4H and 7H. Meanwhile, 5H and 6H performed distinct Oligo-18S rDNA signals on short arm (Appendix A). The distribution of hybridization patterns in these diploid *Hordeum* species differs from that of H genome in *E. dahuricus* detected in our study, indicating that the 18SrDNA-5S rDNA sequences were highly variable from diploid to polyploid of H genome. Meanwhile, the ND-FISH hybridization patterns of Oligo-5S rDNA shown a wide difference with predicted chromosomes, such as 2H chromosome (Appendix A), which possibly implies that the sequence assembly of the related region in barley needs to be improved.

As an essential central component in kinetochore formation, the centromere-specific histone H3 variant (CENH3) has been shown to direct chromosome segregation correctly during mitosis and meiosis [37]. During the evolutionary process of allopolyploidization, the distinctive centromeres from divergent species could affect the assembly of kinetochore proteins [38]. Zhao et al. [39] provided the evidence to show that centromeric specific repeats were well co-localized with CENH3 in octoploid wheat-*Thinopyrum ponticum* partial amphiploids. In present study, all H, St and Y subgenomes displayed chromosomal distribution of CENH3 in the centromeric regions of *E. dahuricus* and barley, and CENH3 signals in H genome were co-localized well with Oligo-HvCSR signals (Figure 6i).

Sequential ND-FISH with multiple probes followed CENH3 immunostaining in the same metaphase was effective to describe the relationship between CENH3 peptide with individual chromosomes. Li et al. [40] found accumulation of CENH3 on centromeric region of wheat-*Thinopyrum* translocation chromosome 1StS.1BL was higher than that of 4BS.4J^S^L and 1BS.1StL chromosomes. Our study distinguished all 42-chromosomes in *E. dahuricus* by different combination of probes, and compare the relative CENH3 fluorescence intensity of each chromosome in H subgenome (Figure 6g). A high variable of CENH3 in 6H chromosome was detected, which implied the existence of the dynamic CENH3 expression among each H chromosome in *E. dahuricus* during evolution of polyploid [31].

Besides centromeric functional regions, centromere-specific tandem repeat sequences are also the essential component in kinetochore formation [41,42]. Centromeric repeats expansion and the formation of new centromere were detected in wheat and its wide hybrid progenies [43]. Meanwhile, Zhao et al. [44] found two distinct CRWs signals on chromosomes 1A and 2A, but only the large domain showed colocalization with CENH3. In our study, ND-FISH by probe Oligo-HvCSR displayed evident signals on centromeric regions of H genome in *E. dahuricus* which are consistent with barley, implying that barley centromeric tandem repeats are conserved within centromeres of H genome between diploid and tetraploid species. Meanwhile, we characterized a rapid increase of relative intensity about Oligo-HvCSR signals in 6H chromosome (Figure 6h). In addition, two centromere repeat sequences-rich regions were also detected on chromosomes 1H, 4H and 4St, but only one region in each chromosome with CENH3 signals (Figure 6i). The results suggested that centromeric repeats of H chromosomes have undergone expansion and tend to insert or transfer to those of St-chromosomes on account of rapid evolutionary rate. However, further verification of centromeric repeats expansion was required precisely quantify techniques for tandem repeats.

Based on the development of long-read and single molecule DNA sequencing technologies, Telomere-to-Telomere (T2T) assembly using deep coverage can substantially improve genome assembly quality [45]. Benefitted from T2T technology, Chen et al. [46] assembled a complete Mo17 genome of maize, with 235 kb thymine- adenine- guanine (TAG) tri-nucleotide repeats and 2,974 copy number of 45S rDNA. As mentioned before, sequence assembly of the 5S rDNA sites in barley need to be largely improved, and further quantitative method of tandem repeats can provide sufficient evidence of centromere expansion in polyploidization of *E. dahuricus*. It is thus to note that the combination of T2T assembly and ND-FISH can be described the rapid changes and determine the accurate copy number of all repetitive regions in wheat and relative species [47], as well as revealing the mechanism of centromere expansion and shift during the processes of polyploidization, speciation and domestication.

## 4. Materials and Methods

### 4.1. Plant Materials

*E. dahuricus* (genomes StStHHYY) was obtained from Chinese Crop Germplasm Information Network. A cultivated barley (*Hordeum vulgare* L.) cv. Morex were maintained in the Laboratory of Molecular and Cell Biology, Center for Informational Biology, School of Life Science and Technology, University of Electronic Science and Technology of China.

### 4.2. Chromosome Preparation and ND-FISH Analysis

Germination of wheat seeds and preparation of mitotic metaphase chromosome from root tips were followed by the description of Han et al. [48]. The protocol of non-denaturing FISH (ND-FISH) using synthesized probes was described by Fu et al. [49]. The sequence of probe Oligo-HvCSR was based on the barley-specific satellite sequence [50]. The synthetic oligonucleotides Oligo-pSc119.2, Oligo-pTa535, Oligo-Po5, Oligo-7E-716, Oligo-7E-599, Oligo-5SrDNA, Oligo-18SrDNA, Oligo-3A1, Oligo-13-J1011, Oligo-7E-744, Oligo-d01-135, Oligo-Ae334 and Oligo-(GAA)_7_ were used for ND-FISH analysis and their sequences are showed in Table 1. All oligonucleotide probes were either 5′ end-labeled with 6-carboxyfluorescein (6-FAM) for green or 6-carboxytetramethylrhodamine (Tamra) for red signals. The pictures of FISH results under Olympus BX-53 microscope were taken by a DP-70 CCD camera (Olympus, Shinjuku, Japan).

### 4.3. GISH and Oligo-FISH Painting

*Pseudoroegneria spicata* genomic DNA for GISH was labeled with digoxigenin-11-dUTP by nick translation according to the manufacturer’s instruction (Roche Diagnostics, Indianapolis, IN, USA). Seven Oligo-FISH pools of probes from Synt1 to Synt7 were used for Oligo-painting [20]. These oligo probes were selected from linkage groups 1 to 7 chromosomes between wheat and barley. The sequential FISH with bulk painting with oligos was performed following the description by Li and Yang [51].

### 4.4. Chromosomal Immunolocalization

The anti-CENH3 antibody was rabbit polyclonal antibody purified by affinity chromatography [52], and was generated by GL Biochem Ltd. (Shanghai, China). Immunolocalization for mitotic metaphase chromosome was performed as described previously [53]. The location of CENH3 was detected with antibody against CENH3 (1:200) and goat anti-rabbit Texas red or green (1:500; Sigma-Aldrich, St Louis, MO, USA). The images were collected with the BX53 Motorized System Microscope (Olympus) and processed using Adobe Photoshop CS 4.0 (Adobe, San Jose, CA, USA).

### 4.5. Drawing Karyotype Ideogram

Ten cells at mitotic metaphase stage were photographed for measurement of arm length and gray value in each *E. dahuricus* chromosomes. The ideogram of St, Y, H chromosomes was conducted by software KaryoMeasure [54]. Software ImageJ was used to process fluorescence in image and to measure the average gray value.

## 5. Conclusions

The present Oligo-FISH painting system visualized homoeologous regions distinctively on wheat and relative species. We have shown that ND-FISH with 14 oligo probes, combined with FISH painting by oligo pools can distinguish the linkage group and sub-genomes of the individual *E. dahuricus* chromosomes. Meanwhile, our current cytogenetic mapping of the novel rearrangement chromosomes integrated with co-localization of CENH3 and sequential ND-FISH, will provide a foundation for future precise chromosome-level genome assembly in *E. dahuricus* in term of polyploidy evolution and environmental adaptation.

## Figures and Tables

**Figure 1 plants-12-03268-f001:**
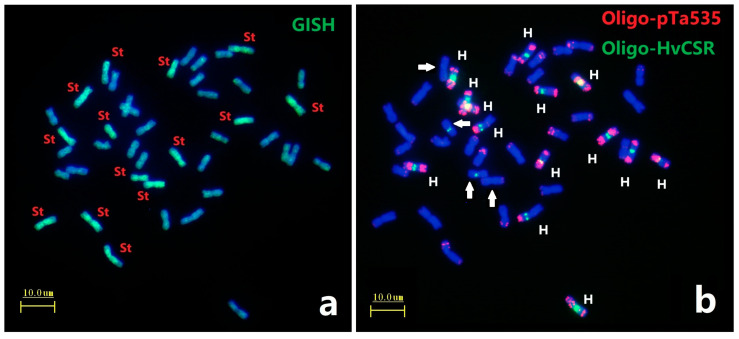
ND-FISH and GISH analyses of metaphase chromosomes of *E. dahuricus*. The GISH probe was total genomic DNA of *Pseudoroegneria spicata* (**a**); ND-FISH probes were Oligo-pTa535 (red) + Oligo-HvCSR (green) (**b**). White arrows indicate the St chromosomes with Oligo-HvCSR signals. Bars, 10 μm.

**Figure 2 plants-12-03268-f002:**
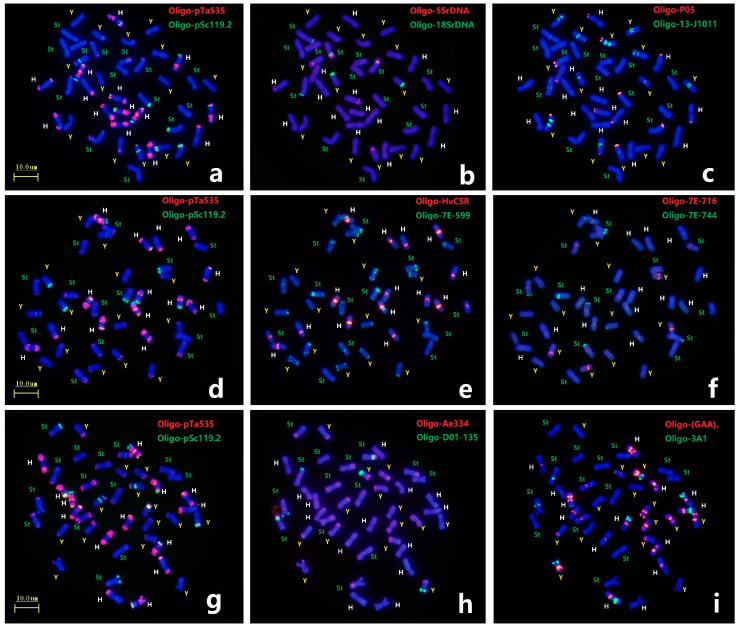
ND-FISH using multiple probes in *E. dahuricus*. The probes for FISH were: Oligo-pTa535 (red) + Oligo-pSc119.2 (green) (**a**,**d**,**g**); Oligo-5SrDNA (red) + Oligo-18SrDNA (green) (**b**); Oligo-T05 (red) + Oligo-13-J1011 (green) (**c**); Oligo-HvCSR (red) + Oligo-7E-599 (green) (**e**); Oligo-7E-716 (red) + Oligo-7E-744 (green) (**f**); Oligo-Ae334 (red) + Oligo-D01-135 (green) (**h**); Oligo-(GAA)_7_ (red) + Oligo-3A1 (green) (**i**). The St, Y and H genomes were marked by green, yellow and white, respectively.

**Figure 3 plants-12-03268-f003:**
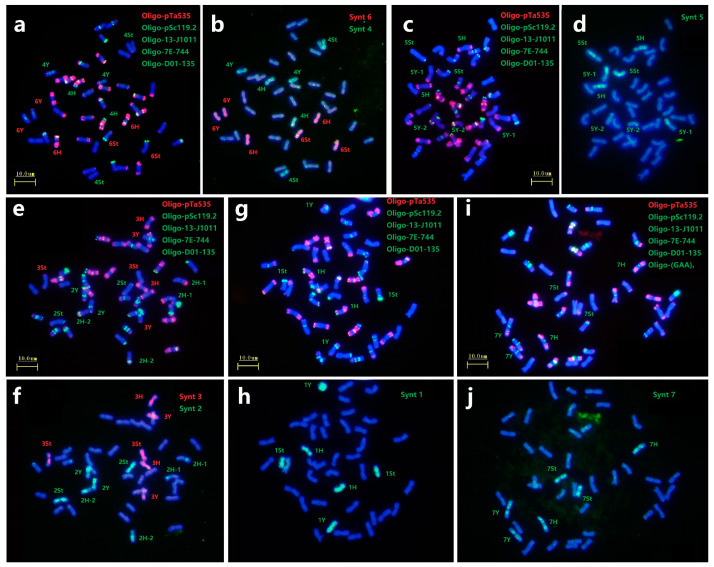
Sequential FISH for *E. dahuricus* with multiple probes and oligo-painting. Probes Oligo-pTa535 + Oligo-pSc119.2 + Oligo-13-J1011 + Oligo-7E-744 + Oligo-D01-135 were used in combined in (**a**,**c**,**e**,**g**), and one more probe Oligo-(GAA)_7_ used in (**i**). Oligo painting probes were Synt4 + Synt6 (**b**), Synt5 (**d**), Synt2 +Synt3 (**f**), Synt1 (**h**), Synt7 (**j**), respectively. The chromosomes were counterstained with DAPI (blue).

**Figure 4 plants-12-03268-f004:**
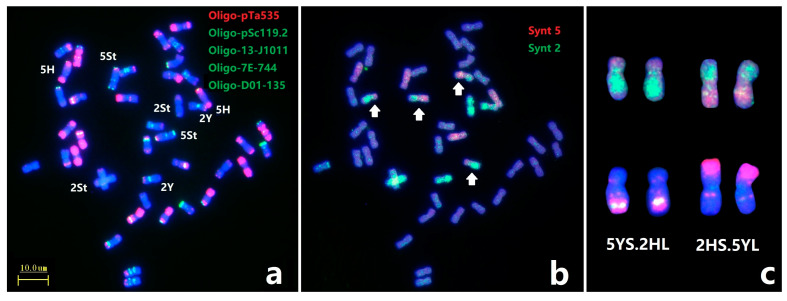
Sequential ND-FISH and Oligo-FISH painting of *E. dahuricus*. The probes for FISH were: Oligo-pTa535 + Oligo-pSc119.2 + Oligo-13-J1011 + Oligo-7E-744 + Oligo-D01-135 (**a**); Oligo-FISH painting probes were: Synt5 (red) + Synt2 (green), white arrow indicate the rearrangement chromosomes (**b**); The ND-FISH and painting karyotypes for 5YS.2HL and 2HS.5YL (**c**).

**Figure 5 plants-12-03268-f005:**
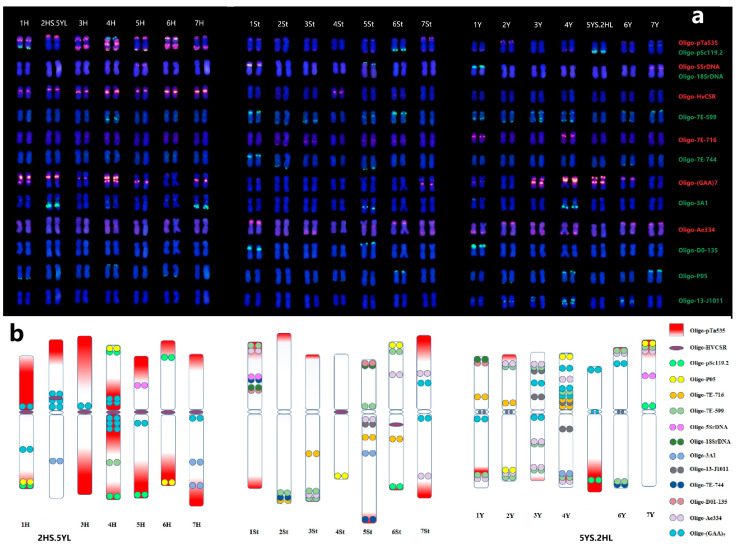
Karyotypes of *E. dahuricus* with homoeologous and subgenome assignment (**a**), the probes list on the right. Ideogram for chromosomes of *E. dahuricus* showing the distribution of multiple probes (**b**).

**Figure 6 plants-12-03268-f006:**
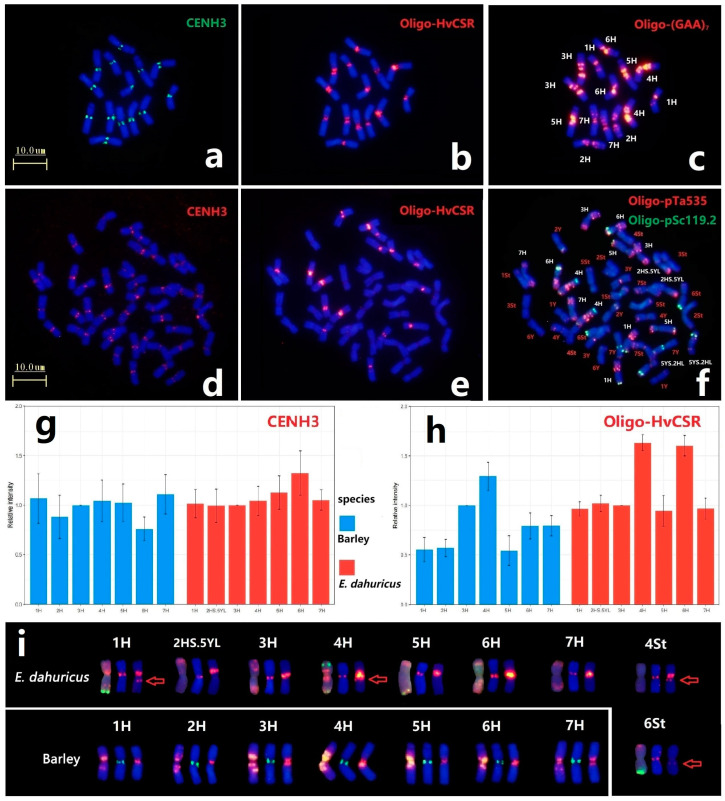
The barley centromeric repeats Oligo-HvCSR and anti-CENH3 location in *E. dahuricus* and barley. The Immunostaining of barley and *E. dahuricus* with anti-CENH3 (**a**,**d**) and sequential ND-FISH of barley by Oligo-HvCSR (**b**) and Oligo-(GAA)_7_ (**c**). ND-FISH of *E. dahuricus* by Oligo-HvCSR (**e**) and Oligo-pTa535 + Oligo-pSc119.2 (**f**). The relative fluorescence intensity of CENH3 and Oligo-HvCSR, compared with 3H chromosome in each metaphase cells (**g**,**h**). The FISH and anti-CENH3 signals of H genome for barley and *E. dahuricus*. (**i**) The compared karyotype of barley and *E. dahuricus* chromosomes from left to right: Oligo-(GAA)_7_ in barley /Oligo-pTa535 + Oligo-pSc119.2 in *E. dahuricus*, CENH3 and Oligo-HvCSR. Red arrow shows the special Oligo-HvCSR more than centromere.

**Table 1 plants-12-03268-t001:** The sequences of oligo probes for *E. dahuricus* chromosome identification by ND-FISH.

Oligo Probes	Sequences	Reference
Oligo-pTa535	AAAAACTTGACGCACGTCACGTACAAATTGGACAAACTCTTTCGGAGTATCAGGGTTTC	[24]
Oligo-pSc119.2	CCGTTTTGTGGACTATTACTCACCGCTTTGGGGTCCCATAGCTAT	[24]
Oligo-5SrDNA	TCAGAACTCCGAAGTTAAGCGTGCTTGGGCGAGAGTAGTAC	[23]
Oligo-18SrDNA	GGGCAAGTCTGGTGCCAGCAGCCGCGGT	[23]
Oligo-7E-744	GCCACCGTGCAGTAGACTTTTTTTGTACCCAAACCATCAGTAACAAAGTTCGTTCAC	[25]
Oligo-3A1	AATAATTTTACACTAGAGTTGAACTAGCTCTATAAGCTAGTTCA	[26]
Oligo-(GAA)_7_	GAAGAAGAAGAAGAAGAAGAA	[27]
Oligo-HvCSR	ACAACGACAACAACGACAATGACGAGA	This study
Oligo-7E-716	GTACAGGACTGCAGCTAAGCCCCCGAGTGAGAGGGTTGCTCATCACTCGGTAGGATT	This study
Oligo-7E-599	CATCGGTCAAACCTCGTCCGGCGAAAGTCAAAGGCGTAGACCGCCCGGTCAACGGTGCC	This study
Oligo-P05	AATACGCTCTTGTTCTTGGCTGTCACGCACATACTTTATGGGATGTCATAGG	This study
Oligo-13-J1011	CATCATGCTTGTTGTGAGAAGCTCTGGTTTGTGAGAAGCATATACCCAAACC	This study
Oligo-D01-135	ACGCGCGCCATGGAAAACAGGGCAAAACCACCGACTCGTCCACGACTCGTAC	This study
Oligo-Ae334	CTCCAAAGTGTTCCTATGGGCTGACCTAACACAACCGGGTGG	This study

## Data Availability

Data are contained within the article.

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
