# Peer review of "Chromosome Rearrangement in Elymus dahuricus Revealed by ND-FISH and Oligo-FISH Painting"

_plants, 2023, doi:10.3390/plants12183268_

Round 1

Reviewer 1 Report

The article addresses an important topic in the field of genomic evolution of polyploids.

The analysis of chromosome rearrangement in Elymus dahuricus was performed with an extensive use of the FISH technique and produced results of considerable interest. However I would like to point out some changes that could improve it. For example, citations that are not strictly necessary should be removed (in the introduction and discussion they are really excessive).  There are also some oversights that have been pointed out in the text.

In my opinion, this article, having made the changes indicated above, can certainly be taken into consideration for publication in this Journal.

Author Response

Citations that are not strictly necessary should be removed (in the introduction and discussion they are really excessive).

Response: Thanks for your suggestion! All revisions including the citations in part of introduction and discussion have been modified accordingly.

Reviewer 2 Report

When describing the role of chromosomal rearrangements the authors should distinguish species-specific rearrangements from others because, in my opinion, unbalanced translocations or random translocations cannot maintain genome stability (line 41-42).

I guess that the term “classification” (instead of “identification”) will be more appropriate (line 66).  

Line 83 – pericentric instead of paracentric.

Figure 1 – white arrows are almost invisible. Please, replace them on the figure.

Please specify Triticeae species whose sequenced genomes were used for design of new oligo-probes (lines 102-103)?

seven pairs of H genome? – line 108.

Lines 108-109 – probably three pairs of 5S and 18S rDNA sites?

Not clear from the text, do author mean three (five, etc.) chromosomes or chromosome pairs? In the case of allogamous species, the odd number of signals is possible; therefore, it is important to clarify the real number of chromosomes carrying hybridization sites.

What is Flexible combination (line 126)?

Please choose different colors for some probes shown on idiogram (Fig. 5). For example, oligo-5S rDNA is difficult to distinguish from Oligo-Ae334.

Several different genomic types are distinguished in barleys, and the H-genome symbol was attributed to the North and South American Hordeum species, while cultivated barley has the I genome. In this study, the authors made a comparison of the H-genome of Elymus with the I genome of barley.  I suggest them to include the results (at least published in literature) on the location of rDNA probe in the discussion section.

Centromere expansion is not equal to centromeric repeat expansion (Discussion part).

Material.

As far as I understood, only one accession of E. dahuricus has been investigated. Owing to this, the reciprocal translocation detected in a study, can present only in this particular accession and therefore cannot be treated as species-specific. Moe materials are needed to confirm specificity of this translocation.

In addition, I really like microphotographs presented in the ms, however they are low informative in their current state. They have too much empty spece, while it was difficult to distinguish signals on chromosomes. I suggest to transfer photos of metaphase cells in supplementary materials and keep only karyotypic images with the sufficient resolution.  Alternatively, if the authors want to keep these illustrations, they should use increase chromosome sizes by reducing empty space. 

Quality of English is not sufficient for publication, text needs significant language edition. Some examples are provided in a previous window. 

Author Response

Introduction:

  1. When describing the role of chromosomal rearrangements the authors should distinguish species-specific rearrangements from others because unbalanced translocations or random translocations cannot maintain genome stability.

Response: We added the detail of species-specific rearrangements in the Introduction section following the suggestion. (lanes 45-48)

  1. The term “classification” (instead of “identification”) will be more appropriate (line 66).

Response: We sincerely thank the reviewer for careful reading. As suggested by the reviewer, we have corrected the “identification” into “classification”.

Results:

  1. The term “pericentric” (instead of “paracentric”) will be more appropriate (line 83).

Response: We have changed the word “paracentric” into “pericentric”.

  1. Figure 1 – White arrows are almost invisible. Please, replace them on the figure.

Response: We modified the white arrows in figure 1 and figure 4.

  1. Please specify Triticeae species whose sequenced genomes were used for design of new oligo-probes (lines 102-103)?

Response: Probe Oligo-HvCSR was designed based on genome of barley cv. Morex; Probe Oligo-7E-716 and Oligo-7E-599 were designed based on the genome of  diploid Thinopyrum elongatum; Probes Oligo-P05, Oligo-13-J1011 and Oligo-D01-135 were designed according to genome of hexaploid Thinopyrum intermedium (https://phytozome-next.jgi.doe.gov/info/ Tintermedium_v2_1); Probe Oligo-Ae334 was developed according to updated version of Aegilops tauschii genome.

  1. Seven pairs of H genome? (line 108).

Response: The correct sentence is “Seven pairs of H chromosomes”.

  1. Probably three pairs of 5S and 18S rDNA sites (Lines 108-109)? Not clear from the text, do author mean three (five, etc.) chromosomes or chromosome pairs? In the case of allogamous species, the odd number of signals is possible; therefore, it is important to clarify the real number of chromosomes carrying hybridization sites.

Response: Thanks for your correction. We agree to edit the description as “E. dahuricus displayed three pairs of 5S and 18S rDNA sites”.

  1. What is Flexible combination (line 126)?

Response: “Flexible combination” was something of a mistake. We agree to change the sentence as “Seven pairs of St chromosomes can be recognized by the different combination of these four probes”.

  1. Please choose different colors for some probes shown on ideogram (Fig 5). For example, oligo-5S rDNA is difficult to distinguish from Oligo-Ae334.

Response: We modified the colors of some probes on ideogram accordingly. For example, we changed color of probe Oligo-Ae334 into purple.

  1. Microphotographs are low informative in their current state. They have too much empty space, while it was difficult to distinguish signals on chromosomes. I suggest to transfer photos of metaphase cells in supplementary materials and keep only karyotypic images with the sufficient resolution. Alternatively, if the authors want to keep these illustrations, they should use increase chromosome sizes by reducing empty space.

Response: Thank you for your reminder. As shown in figure 1-5, we have reduced empty space as far as possible to keep the better resolution of each image.

Discussion

  1. Several different genomic types are distinguished in barleys, and the H-genome symbol was attributed to the North and South American Hordeum species, while cultivated barley has the I genome. In this study, the authors made a comparison of the H-genome of Elymus with the I genome of barley. I suggest them to include the results (at least published in literature) on the location of rDNA probe in the discussion section.

Response: Thanks for your suggestion. We added the location of rDNA probes in different Hordeum species with I-, H-, X- and Y-genome species, and made a comparison of those genome with H-genome of E. dahuricus in rDNA hybridization sites (Line 311-318).

  1. Centromere expansion is not equal to centromeric repeat expansion.

Response: In the discussion section, “centromere expansion” in line 341 and line 354 were changed into “centromeric repeats expansion”.

Material

  1. As far as I understood, only one accession of dahuricus has been investigated. Owing to this, the reciprocal translocation detected in a study, can present only in this particular accession and therefore cannot be treated as species-specific. More materials are needed to confirm specificity of this translocation.

Response: We sincerely thank the reviewer for careful reading. The main reasons that we would like to consider treating the reciprocal translocation detected in our study as species-specific CRs are showed as following: 1. Yang et al. (2017) investigated the genomic constitution and intergenomic translocations in the Elymus dahuricus complex by multicolor GISH, and the results showed that 7H (shortest chromosome in H genome) and 1Y (longest chromosome in Y genome) involved in reciprocal translocations for all the accessions. 2. Our study noted that only one reciprocal translocation occurred between H and Y genome. By the way of chromosomal length measurement, 2HS.5YL chromosome performed longest in Y genome corresponding 1Y (longest chromosome in Y genome) described by Yang et al. The same as 5YS.2HL chromosome, was taken for 7H (shortest chromosome in H genome). Thus, we have sufficient reasons to consider that the CRs of E. dahuricus detected in our study are species-specific CRs. According to your suggestions, we will take more materials into research to confirm species-specific CRs in later work.

  1. Quality of English is not sufficient for publication, text needs significant language edition. Some examples are provided in a previous window.

Response: Thanks for your suggestion. We will invite an expert of a native English speaking to polish our article in the revised paper. We hope the revised manuscript could be acceptable for you.